# SPARSE Q-LEARNING: OFFLINE REINFORCEMENT LEARNING WITH IMPLICIT VALUE REGULARIZATION

## ABSTRACT

Most offline reinforcement learning (RL) methods suffer from the trade-off between improving the policy to surpass the behavior policy and constraining the policy to limit the deviation from the behavior policy as computing Q-values using out-of-distribution actions will suffer from errors due to distributional shift. The recent proposed *In-sample Learning* paradigm (e.g., IQL), which improves the policy by quantile regression using only data samples, shows great promise because it learns an optimal policy without querying the value function of any unseen actions. However, it remains unclear how this type of method handles the distributional shift in learning the value function. In this work, we make a key finding that the in-sample learning paradigm arises under the *Implicit Value Regularization* (IVR) framework. This gives a deeper understanding of why the in-sample learning paradigm works, i.e., it applies implicit value regularization to the policy. Based on the IVR framework, we further propose a practical algorithm, which uses the same value regularization as CQL, but in a complete in-sample manner. Compared with IQL, we find that our algorithm introduces sparsity in learning the value function, we thus dub our method Sparse Q-learning (SQL). We verify the effectiveness of SQL on D4RL benchmark datasets. We also show the benefits of sparsity by comparing SQL with IQL in noisy data regimes and show the robustness of in-sample learning by comparing SQL with CQL in small data regimes. Under all settings, SQL achieves better results and owns faster convergence compared to other baselines.

## 1 INTRODUCTION

Reinforcement learning (RL) is an increasingly important technology for developing highly capable AI systems, it has achieved great success in game-playing domains (Mnih et al., 2013; Silver et al., 2017). However, the fundamental online learning paradigm in RL is also one of the biggest obstacles to RL's widespread adoption, as interacting with the environment can be costly and dangerous in real-world settings. Offline RL, also known as batch RL, aims at solving the abovementioned problem by learning effective policies solely from offline data, without any additional online interactions. It is a promising area for bringing RL into real-world domains, such as robotics (Kalashnikov et al., 2021), healthcare (Tang & Wiens, 2021) and industrial control (Zhan et al., 2022). In such scenarios, arbitrary exploration with untrained policies is costly or dangerous, but sufficient prior data is available.

While most off-policy RL algorithms are applicable in the offline setting by filling the replay buffer with offline data, improving the policy beyond the level of the behavior policy entails querying the Q-function about values of actions produced by the policy, which are often not seen in the dataset. Those out-of-distribution actions can be deemed as adversarial examples of the Q-function, which cause extrapolation errors of the Q-function (Kumar et al., 2020). To alleviate this issue, prior model-free offline RL methods typically add pessimism to the learning objective, in order to be pessimistic about the distributional shift. Pessimism can be achieved by policy constraint, which constrains the policy to be close to the behavior policy (Kumar et al., 2019; Wu et al., 2019; Nair et al., 2020; Fujimoto & Gu, 2021); or value regularization, which directly modifies the Q-function to be pessimistic (Kumar et al., 2020; Kostrikov et al., 2021a; An et al., 2021; Bai et al., 2021). Nevertheless, this imposes a trade-off between accurate value estimation (more regularization) and maximum policy performance (less regularization).

In this work, we find that we could alleviate the trade-off in *out-of-sample learning* by performing *implicit value regulazization*, this bypasses querying the value function of any unseen actions, allows learning an optimal policy using *in-sample learning*[1]. More specifically, we propose the Implicit Value Regulazization (IVR) framework, in which a general form of behavior regularizers is added to the policy learning objective. Because of the regularization, the optimal policy in the IVR framework has a closed-form solution, which can be expressed by imposing weight on the behavior policy. The weight can be computed by a state-value function and an action-value function, the state-value function serves as a normalization term to make the optimal policy integrate to 1. It is usually intractable to find a closed form of the state-value function, however, we make a subtle mathematical transformation and show its equivalence to solving a convex optimization problem. In this manner, both of these two value functions can be learned by only dataset samples.

Note that the recently proposed method, IQL (Kostrikov et al., 2021b), also learns a state-value function and an action-value function iteratively, then extracts the policy using these two value functions. Although derived from a different view (i.e., approximate an upper expectile of dataset actions given a state), IQL remains much close to the learning paradigm of our framework. Furthermore, our IVR framework explains why learning the state-value function is important in IQL and gives a deeper understanding of how IQL handles the distributional shift: it is doing implicit value regularization, with the hyperparameter $\tau$ to control the strength. This explains one disturbing issue of IQL, i.e., the role of $\tau$ does not have a perfect match between theory and practice. In theory, $\tau$ should be close to 1 to obtain an optimal policy while in practice a larger $\tau$ may give a worse result.

Based on the IVR framework, we further propose a practical algorithm. We find that the value regularization term used in CQL belongs to one of the valid choices in our framework. However, when applying it to our framework, we get a complete in-sample learning algorithm. The resulting algorithm also bears similarities to IQL, we find that our algorithm introduces sparsity in learning the state-value function, which is missing in IQL. The sparsity term filters out those bad actions whose Q-values are below a threshold, which brings benefits when the quality of offline datasets is inferior, we thus dub our method Sparse Q-learning (SQL). We verify the effectiveness of SQL on widely-used D4RL benchmark datasets and demonstrate the state-of-the-art performance, especially on suboptimal datasets in which value learning is necessary (e.g., Antmaze and Kitchen). We also show the benefits of sparsity by comparing SQL with IQL in noisy data regimes and the robustness of in-sample learning by comparing SQL with CQL in small data regimes. Under all settings, SQL achieves better results and owns faster convergence compared with other baselines.

To summarize, the contributions of this paper are as follows:

- We propose a general implicit value regularization framework, where different behavior regularizers can be included, all leading to a complete in-sample learning paradigm.

- Based on the proposed framework, we design an effective offline RL algorithm, Sparse Q-Learning, which obtains SOTA results on benchmark datasets and shows robustness in both noisy and small data regimes.

## 2 RELATED WORK

In order to tackle the distributional shift problem, most model-free offline RL methods augment existing off-policy methods (e.g., Q-learning or actor-critic) with a behavior regularization term. The primary ingredient of this class of methods is to propose various regularizers to ensure that the learned policy does not stray too far from the behavior policy, i.e., stays in distribution. These regularizers can appear explicitly as divergence penalties (Wu et al., 2019; Kumar et al., 2019; Fujimoto & Gu, 2021), implicitly through weighted behavior cloning (Wang et al., 2020; Nair et al., 2020), or more directly through careful parameterization of the policy (Fujimoto et al., 2018; Zhou et al., 2020). Another way to apply behavior regularizers is via modification of the critic learning objective to incorporate some form of regularization, so as to encourage staying near the behavioral distribution and being pessimistic about unknown state-action pairs (Nachum et al., 2019; Kumar et al., 2020; Kostrikov et al., 2021a; Xu et al., 2022). There are also several works incorporating behavior regularization

---

[1]The core difference between in-sample learning and out-of-sample learning is that in-sample learning uses only dataset actions to learn the value function while out-of-sample learning uses actions produced by the policy.

through the use of uncertainty (Wu et al., 2021; An et al., 2021; Bai et al., 2021) or distance function (Dadashi et al., 2021; Li et al., 2022).

However, in-distribution constraints used in these works might not be sufficient to avoid value function extrapolation errors. Another line of methods, on the contrary, avoid value function extrapolation by performing some kind of imitation learning on the dataset. When the dataset is good enough or contains high-performing trajectories, we can simply clone or filter dataset actions to extract useful transitions. For instance, recent work filters trajectories based on their return (Chen et al., 2020; Peng et al., 2019), or directly filters individual transitions based on how advantageous they could be under the behavior policy and then clones them Brandfonbrener et al. (2021); Gulcehre et al. (2021). While alleviating extrapolation errors, these methods only perform single-step dynamic programming, and lose the ability to "stitch" suboptimal trajectories by multi-step dynamic programming.

Our method can be viewed as a combination of these two methods while sharing the best of both worlds: SQL implicitly controls the distributional shift, and learns an optimal policy by in-sample generalization. SQL is less vulnerable to erroneous value estimation as in-sample actions induce less distributional shift than out-of-sample actions. Similar to our work, IQL (Kostrikov et al., 2021b) approximates the optimum by fitting the upper expectile of the behavior policy's action-value function, however, it is not motivated by remaining pessimistic to the distributional shift.

Our method adds a behavior regularization term to the RL learning objective. In online RL, there is also some work incorporating an entropy-regularized term into the learning objective (Haarnoja et al., 2018; Nachum et al., 2017; Lee et al., 2019; Neu et al., 2017; Geist et al., 2019; Ahmed et al., 2019), this brings multi-modality to the policy and is beneficial for the exploration. Note that the entropy-regularized term only involves the policy, it could be directly computed, resulting in a similar learning procedure as in SAC (Haarnoja et al., 2018). While our method considers the offline setting, and provides a different learning procedure to solve the problem by jointly learning a state-value function and an action-value function.

## 3 PRELIMINARIES

We consider the RL problem presented as a Markov Decision Process (MDP) (Sutton et al., 1998), which is specified by a tuple $\mathcal{M} = \langle \mathcal{S}, \mathcal{A}, T, r, \rho, \gamma \rangle$ consisting of a state space, an action space, a transition probability function, a reward function, an initial state distribution, and the discount factor. The goal of RL is to find a policy $\pi(a|s) : \mathcal{S} \times \mathcal{A} \to [0, 1]$ that maximizes the expected discounted cumulative reward (or called return) along a trajectory as

$$\max_\pi \mathbb{E}\left[\sum_{t=0}^\infty \gamma^t r\left(s_t, a_t\right) \middle| s_0 = s, a_0 = a, s_t \sim T\left(\cdot|s_{t-1}, a_{t-1}\right), a_t \sim \pi\left(\cdot|s_t\right) \text{ for } t \geq 1\right]. \quad (1)$$

In this work, we focus on the offline setting. Unlike online RL methods, offline RL aims to learn an optimal policy from a fixed dataset $\mathcal{D}$ consisting of trajectories that are collected by different policies. The dataset can be heterogenous and suboptimal, we denote the underlying behavior policy of $\mathcal{D}$ as $\mu$, which represents the conditional distribution $p(a|s)$ observed in the dataset.

RL methods based on approximate dynamic programming (both online and offline) typically maintain an action-value function ($Q$-function) and, optionally, a state-value function ($V$-function), refered as $Q(s, a)$ and $V(s)$ respectively (Haarnoja et al., 2017; Nachum et al., 2017; Kumar et al., 2020; Kostrikov et al., 2021b). These two value functions are learned by encouraging them to satisfy single-step Bellman consistencies. Define a collection of policy evaluation operator (of different policy $\mathbf{x}$) on $Q$ and $V$ as

$$(\mathcal{T}^{\mathbf{x}} Q)(s, a) := r(s, a) + \gamma \mathbb{E}_{s'|s,a} \mathbb{E}_{a' \sim \mathbf{x}} \left[Q(s', a')\right]$$

$$(\mathcal{T}^{\mathbf{x}} V)(s) := \mathbb{E}_{a \sim \pi} \left[r(s, a) - \gamma \mathbb{E}_{s'|s,a} \left[V(s')\right]\right],$$

then $Q$ and $V$ are learned by $\min_Q J(Q) = \frac{1}{2} \mathbb{E}_{(s,a)\sim\mathcal{D}} \left[(\mathcal{T}^{\mathbf{x}} Q - Q)(s, a)^2\right]$ and $\min_V J(V) = \frac{1}{2} \mathbb{E}_{s\sim\mathcal{D}} \left[(\mathcal{T}^{\mathbf{x}} V - V)(s)^2\right]$, respectively. Note that $\mathbf{x}$ could be the learned policy $\pi$ or the behavior policy $\mu$, if $\mathbf{x} = \mu$, then $a \sim \mu$ and $a' \sim \mu$ are equal to $a \sim \mathcal{D}$ and $a' \sim \mathcal{D}$, respectively. In offline RL, since $\mathcal{D}$ typically does not contain all possible transitions $(s, a, s')$, one actually uses an empirical policy evaluation operator that only backs up a single $s'$ sample, we denote this operator as $\hat{\mathcal{T}}^{\mathbf{x}}$.

**In-sample Learning via Expectile Regression**   Instead of adding explicit regularization to the policy evaluation operator to avoid out-of-distribution actions, IQL uses only in-sample actions to learn the optimal $Q$-function. IQL uses an asymmetric $\ell_2$ loss (i.e., expectile regression) to learn the $V$-function, which can be seen as an estimate of the maximum $Q$-value over actions that are in the dataset support, thus allowing implicit Q-learning:

$$\min_V \ \mathbb{E}_{(s,a)\sim\mathcal{D}}\big[\big|\tau - \mathbb{1}\big(Q(s,a) - V(s) < 0\big)\big|\big(Q(s,a) - V(s)\big)^2\big] \tag{2}$$

$$\min_Q \ \mathbb{E}_{(s,a,s')\sim\mathcal{D}}\big[\big(r(s,a) + \gamma V(s') - Q(s,a)\big)^2\big],$$

where $\mathbb{1}$ is the indicator function. After learning $Q$ and $V$, IQL extracts the policy by advantage-weighted regression (Peters et al., 2010; Peng et al., 2019; Nair et al., 2020):

$$\min_\pi \ \mathbb{E}_{(s,a)\sim\mathcal{D}}\big[e^{(Q(s,a)-V(s))/\alpha}\log\pi(a|s)\big]. \tag{3}$$

While IQL achieves superior D4RL benchmark results, several issues remain unsolved:

- The hyperparameter $\tau$ has a gap between theory and practice: in theory $\tau$ should be close to 1 to obtain an optimal policy while in practice a larger $\tau$ may give a worse result.
- In IQL the value function is estimating the optimal policy instead of the behavior policy, how does IQL handle the distributional shift issue?
- Why should the policy be extracted by advantage-weighted regression, does this technique guarantee the same optimal policy as the one implied in the learned optimal $Q$-function?

## 4   OFFLINE RL WITH IMPLICIT VALUE REGULARIZATION

In this section, we introduce a framework where a general form of value regularization can be implicitly applied. We begin with a special MDP where a behavior regularizer is added to the reward, we conduct a full mathematical analysis of this regularized MDP and give the solution of it under certain assumptions, which results in a complete in-sample learning paradigm. We then instantiate a practical algorithm from this framework and give a thorough analysis and discussion of it.

### 4.1   BEHAVIOR-REGULARIZED MDPS

Like entropy-regularized RL adds a entropy regularizer to the reward (Haarnoja et al., 2018), in this paper we consider imposing a general behavior regularization term to objective (1) and solve the following *behavior-regularized* MDP problem

$$\max_\pi \ \mathbb{E}\bigg[\sum_{t=0}^\infty \gamma^t\Big(r(s_t, a_t) - \alpha \cdot f\Big(\frac{\pi(a_t|s_t)}{\mu(a_t|s_t)}\Big)\Big)\bigg], \tag{4}$$

where $f(\cdot)$ is a regularization function. It is known that in entropy-regularized RL the regularization gives smoothness of the Bellman operator (Ahmed et al., 2019; Chow et al., 2018), e.g., from greedy max to softmax over the whole action space when the regularization is Shannon entropy. While in our new learning objective (4), we find that the smoothness will transfer the greedy max from policy $\pi$ to a softened max (depending on $f$) over behavior policy $\mu$, this enables an in-sample learning scheme, which is appealing in the offline RL setting.

In the behavior-regularized MDP, we have a modified policy evaluation operator $\mathcal{T}_f^\pi$ given by

$$(\mathcal{T}_f^\pi)Q(s,a) := r(s,a) + \gamma\mathbb{E}_{s'|s,a}\big[V(s')\big]$$

where

$$V(s) = \mathbb{E}_{a\sim\pi}\bigg[Q(s,a) - \alpha f\Big(\frac{\pi(a|s)}{\mu(a|s)}\Big)\bigg].$$

The policy learning objective can also be expressed as $\max_\pi \mathbb{E}_{s\sim\mathcal{D}}\big[V(s)\big]$. Compared with the origin policy evaluation operator $\mathcal{T}^\pi$, $\mathcal{T}_f^\pi$ is actually applying a value regularization to the $Q$-function. However, the regularization term is hard to compute because the behavior policy $\mu$ is unknown. Although we can use Fenchel-duality (Boyd et al., 2004) to get a sampled-based estimation if $f$ belongs to the $f$-divergence (Wu et al., 2019), this unnecessarily brings a min-max optimization problem, which is hard to solve and results in a poor performance in practice (Nachum et al., 2019).

## 4.2 ASSUMPTIONS AND SOLUTIONS

We now show that we can get the optimal value function $Q^*$ and $V^*$ without knowing $\mu$. First, in order to make the learning problems (4) analyzable, two basic assumptions are required as follows:

**Assumption 1.** *Assume $\pi(a|s) > 0 \Rightarrow \mu(a|s) > 0$ so that $\pi/\mu$ is well-defined.*

**Assumption 2.** *Assume the function $f(x)$ satisfies the following conditions on $(0, \infty)$ : (1) $f(1) = 0$; (2) $h_f(x) = xf(x)$ is strictly convex; (3) $f(x)$ is differentiable.*

The assumptions of $f(1) = 0$ and $xf(x)$ strictly convex make the regularization term be positive due to the Jensen's inequality as $\mathbb{E}_\mu \left[ \frac{\pi}{\mu} f \left( \frac{\pi}{\mu} \right) \right] \geq 1 f(1) = 0$. This guarantees that the regularization term is minimized only when $\pi = \mu$. Because $h_f(x)$ is strictly convex, its derivative, $h'_f(x) = f(x) + xf'(x)$ is a strictly increasing function and thus $(h'_f)^{-1}(x)$ exists. For simplicity, we denote $g_f(x) = (h'_f)^{-1}(x)$. The assumption of differentiability facilitates theoretic analysis and benefits practical implementation due to the widely used automatic derivation in deep learning.

Under these two assumptions, we can get the following two theorems:

**Theorem 1.** *In the behavior-regularized MDP, any optimal policy $\pi^*$ and its optimal value function $Q^*$ and $V^*$ satisfy the following optimality condition for all states and actions:*

$$Q^*(s, a) = r(s, a) + \gamma \mathbb{E}_{s'|s,a} [V^*(s')]$$

$$\pi^*(a|s) = \mu(a|s) \cdot \max \left\{ g_f \left( \frac{Q^*(s,a) - U^*(s)}{\alpha} \right), 0 \right\} \tag{5}$$

$$V^*(s) = U^*(s) + \alpha \mathbb{E}_{a \sim \mu} \left[ \left( \frac{\pi^*(a|s)}{\mu(a|s)} \right)^2 f' \left( \frac{\pi^*(a|s)}{\mu(a|s)} \right) \right] \tag{6}$$

*where $U^*(s)$ is a normalization term so that $\sum_{a \in \mathcal{A}} \pi^*(a|s) = 1$.*

The proof is provided in Appendix 7.1. The proof depends on the KKT condition where the derivative of a Lagrangian objective function with respect to policy $\pi(a|s)$ becomes zero at the optimal solution. Note that the resulting formulation of $Q^*$ and $V^*$ only involves $U^*$ and action samples from $\mu$. $U^*(s)$ can be uniquely solved from the equation obtained by plugging Eq.(5) into $\sum_{a \in \mathcal{A}} \pi^*(a|s) = 1$, which also only uses actions sampled from $\mu$. In other words, now the learning of $Q^*$ and $V^*$ can be realized in an in-sample manner.

Theorem 1 also shows how the behavior regularization influences the optimality condition. If we choose $f$ such that there exists some $x$ that $g_f(x) < 0$, then it can be shown from Eq.(5) that the optimal policy $\pi^*$ will be sparse by assigning zero probability to the actions whose $Q$-values $Q^*(s, a)$ are below the threshold $U^*(s) + \alpha h'_f(0)$ and assigns positive probability to near optimal actions in proportion to their $Q$-values (since $g_f(x)$ is increasing). Note that $\pi^*$ could also have no sparsity, for example, if we choose $f = \log(x)$, then $g_f = \exp(x)$ will give all elements non-zero values.

**Theorem 2.** *Define $\mathcal{T}_f^*$ the case where $\pi$ in $\mathcal{T}_f^\pi$ is the optimal policy $\pi^*$, then $\mathcal{T}_f^*$ is a $\gamma$-contraction.*

The proof is provided in Appendix 7.2. This theorem means that by applying $Q^{k+1} = \mathcal{T}_f^* Q^k$ repeatedly, then sequence $Q^k$ will converge to the $Q$-value of the optimal policy $\pi^*$ when $k \to \infty$.

## 4.3 SPARSE Q-LEARNING

After giving the closed-form solution of the optimal value function. We now aim to instantiate a practical algorithm. In offline RL, in order to completely avoid out-of-distribution actions, we want a *zero-forcing* support constraint, i.e., $\mu(a|s) = 0 \Rightarrow \pi(a|s) = 0$. This reminds us of the class of $\alpha$-divergence (Boyd et al., 2004), which is a subset of $f$-divergence and takes the following form ($\alpha \in \mathbb{R} \backslash \{0, 1\}$):

$$D_\alpha(\mu, \pi) = \frac{1}{\alpha(\alpha - 1)} \mathbb{E}_\pi \left[ \left( \frac{\pi}{\mu} \right)^{-\alpha} - 1 \right].$$

$\alpha$-divergence is known to be mode-seeking if one chooses $\alpha \leq 0$. Note that the Reverse KL divergence is the limit of $D_\alpha(\mu, \pi)$ when $\alpha \to 0$. We can also obtain Helinger distance and Neyman

$\chi^2$-divergence as $\alpha = 1/2$ and $\alpha = -1$, respectively. One interesting property of $\alpha$-divergence is that $D_\alpha(\mu, \pi) = D_{1-\alpha}(\pi, \mu)$.

Note that CQL (Kumar et al., 2020) applies the following conservative policy evaluation operator to learn the $Q$-function (according to Appendix C, equation 9):

$$Q(s, a) = \mathcal{T}^\pi Q(s, a) - \beta \left[ \frac{\pi(\mathbf{a}|\mathbf{s})}{\mu(\mathbf{a}|\mathbf{s})} - 1 \right],$$

which is the same as $\alpha = -1$. Hence, CQL is implicitly doing value regularization by applying the Neyman $\chi^2$-divergence.

In this work, we consider the same behavior regularizer, which means $f(x) = x - 1$ and $g_f(x) = \frac{1}{2}x + \frac{1}{2}$. Plug them into Eq.(5) and Eq.(6) in Theorem 1, we get the following formulation:

$$Q^*(s, a) = r(s, a) + \gamma \mathbb{E}_{s'|s,a}\left[ V^*(s') \right] \tag{7}$$

$$\pi^*(a|s) = \mu(a|s) \cdot \max\left\{ \frac{1}{2} + \frac{Q^*(s, a) - U^*(s)}{2\alpha}, 0 \right\} \tag{8}$$

$$V^*(s) = U^*(s) + \alpha \mathbb{E}_{a\sim\mu}\left[ \left( \frac{\pi^*(a|s)}{\mu(a|s)} \right)^2 \right], \tag{9}$$

where $U^*(s)$ needs to satisfy the following equation to make $\pi^*$ integrate to 1:

$$\mathbb{E}_{a\sim\mu}\left[ \max\left\{ \frac{1}{2} + \frac{Q^*(s, a) - U^*(s)}{2\alpha}, 0 \right\} \right] = 1 \tag{10}$$

It is usually intractable to get the closed-form solution of $U^*(s)$ from Eq.(10), however, here we make a mathematical transformation and show its equivalence to solving a convex optimization problem.

**Lemma 1.** *We can get $U^*(s)$ by solving the following optimization problem:*

$$\min_U \mathbb{E}_{a\sim\mu}\left[ \mathbb{1}\left( \frac{1}{2} + \frac{Q^*(s, a) - U(s)}{2\alpha} > 0 \right) \left( \frac{1}{2} + \frac{Q^*(s, a) - U(s)}{2\alpha} \right)^2 \right] + \frac{U(s)}{\alpha} \tag{11}$$

The proof can be easily got if we set the derivative of the objective to 0 with respect to $U^*(s)$, which is exactly Eq.(10). Now we obtain a learning scheme to get $Q^*$, $U^*$ and $V^*$ by iteratively updating $Q$, $U$ and $V$ following Eq.(9), objective (11) and Eq.(7), respectively. We refer to this learning scheme as SQL-U, however, SQL-U needs to train three networks, which is a bit computationally expensive.

Note that the term $\mathbb{E}_{a\sim\mu}\left[ \left( \frac{\pi^*(a|s)}{\mu(a|s)} \right)^2 \right]$ in Eq.(9) is equal to $\mathbb{E}_{a\sim\pi^*}\left[ \frac{\pi^*(a|s)}{\mu(a|s)} \right]$, as $\pi^*$ is optimized to become mode-seeking, for actions sampled from $\pi^*$, its probability $\pi^*(a|s)$ should be close to the probability under the behavior policy, $\mu^*(a|s)$. Note that for actions sampled from $\mu$, $\pi^*(a|s)$ and $\mu^*(a|s)$ may have a large difference because $\pi^*(a|s)$ may be 0.

Hence in SQL we make an approximation by assuming $\mathbb{E}_{a\sim\pi^*}\left[ \frac{\pi^*(a|s)}{\mu(a|s)} \right] = 1$, this removes one network as $U^* = V^* - \alpha$. Replacing $U^*$ with $V^*$, we get the following learning scheme that only needs to learn $V$ and $Q$ iteratively to get $V^*$ and $Q^*$:

$$\min_V \mathbb{E}_{(s,a)\sim\mathcal{D}}\left[ \mathbb{1}\left( 1 + \frac{Q(s, a) - V(s)}{2\alpha} > 0 \right) \left( 1 + \frac{Q(s, a) - V(s)}{2\alpha} \right)^2 + \frac{V(s)}{\alpha} \right] \tag{12}$$

$$\min_Q \mathbb{E}_{(s,a,s')\sim\mathcal{D}}\left[ \left( r(s, a) + \gamma V(s') - Q(s, a) \right)^2 \right] \tag{13}$$

After getting $V$ and $Q$, following the formulation of $\pi^*$ in Eq.(14), we can get the learning objective of policy $\pi$ by minimizing the KL-divergence between $\pi$ and $\pi^*$ (Haarnoja et al., 2018):

$$\min_\pi \mathbb{E}_{(s,a)\sim\mathcal{D}}\left[ \mathbb{1}\left( 1 + \frac{Q(s, a) - V(s)}{2\alpha} > 0 \right) \left( 1 + \frac{Q(s, a) - V(s)}{2\alpha} \right) \log \pi(a|s) \right]. \tag{14}$$

Our final algorithm, SQL, consists of three supervised stages: learning $V$, learning $Q$, and learning $\pi$. We use target networks for $Q$-functions and use clipped double Q-learning (take the minimum of two $Q$-functions) for $V$ and $\pi$ updates. We summarize the training procedure of SQL in Algorithm 1.

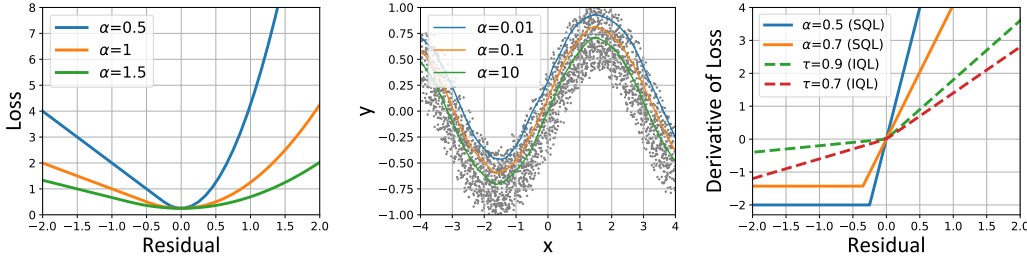

Figure 1: **Left:** The loss with respect to the residual ($Q - V$) in the learning objective of $V$ in SQL with different $\alpha$. **Center:** An example of estimating state conditional extrema of a two-dimensional random variable (generated by adding random noise to samples from $y = \sin(x)$). Each $x$ corresponds to a distribution over $y$. The loss fits the extrema more with $\alpha$ becoming smaller. **Right:** The comparison of the derivative of loss of SQL and IQL. In SQL, the derivative keeps unchanged when the residual is below a threshold.

## 4.4 DISCUSSIONS

**A statistical view of why SQL work**    Inspired by the analysis in IQL, we give another view of why SQL could learn the optimal policy. Consider estimating a parameter $m_\alpha$ for a random variable $X$ using samples from a dataset $\mathcal{D}$, we show that $m_\alpha$ could fit the extrema of $X$ by using the learning objective of $V$-function in SQL:

$$\arg \min_{m_\alpha} \; \mathbb{E}_{x \sim \mathcal{D}} \left[ \mathbb{1}\left(1 + \frac{x - m_\alpha}{2\alpha} > 0\right)\left(1 + \frac{x - m_\alpha}{2\alpha}\right)^2 + \frac{m_\alpha}{\alpha} \right]$$

In Figure 1, we give an example of estimating the state conditional extrema of a two-dimensional random variable, as shown, $\alpha \to 0$ approximates the maximum operator over in-support values of $y$ given $x$. This phenomenon can be justified in our IVR framework as the value function becomes more optimal with less value regularization. However, less value regularization also brings more distributional shift, so we need a proper $\alpha$ to trade-off optimality with distributional shift.

**Connections to prior works**    SQL establishes the connection with several prior works such as CQL (Kumar et al., 2020), IQL (Kostrikov et al., 2021b) and OptiDICE (Lee et al., 2021).

Like CQL pushes down policy $Q$-values and pushes up dataset $Q$-values, in SQL, the first term in Eq.(12) pushes up $V$-values if $Q - V > 0$ while the second term pushes down $V$-values, and $\alpha$ trades these two terms off. SQL incorporates the same inherent conservatism as CQL by adding the $\chi^2$-divergence to the policy evaluation operator. However, SQL learns the value function

---

**Algorithm 1** Sparse Q-Learning (SQL)

**Require:** $\mathcal{D}, \alpha$.
1: Initialize $Q_\phi, Q_{\phi'}, V_\psi, \pi_\theta$
2: **for** $t = 1, 2, \cdots, N$ **do**
3:     Sample transitions $(s, a, r, s') \sim \mathcal{D}$
4:     Update $V_\psi$ by Eq.(12) using $V_\psi, Q_{\phi'}$
5:     Update $Q_\phi$ by Eq.(13) using $V_\psi, Q_\phi$
6:     Update $Q_{\phi'}$ by $\phi' \leftarrow \lambda\phi + (1 - \lambda)\phi'$
7:     Update $\pi_\theta$ by Eq.(14) using $V_\psi, Q_{\phi'}$
8: **end for**

---

using only dataset samples while CQL needs to sample actions from the policy. In this sense, SQL is an "implicit" version of CQL that avoids any out-of-distribution action.

Like IQL, SQL learns both $V$-function and $Q$-function. However, IQL seems to be a heuristic approach and the learning objective of $V$-function in IQL has a drawback. We compute the derivative of the $V$-function learning objective with respect to the residual ($Q - V$) in SQL and IQL, it can be seen in Figure 1 right that SQL keeps the derivative unchanged when the residual is below a threshold, while IQL doesn't. In IQL, the derivative keeps decreasing as the residual becomes more negative, hence, the $V$-function will be over-underestimated by those bad actions whose $Q$-value is extremely small. Note that SQL will assign a zero probability mass to those bad actions according to Eq.(14), the sparsity is incorporated due to the mode-seeking behavior of $\chi^2$-divergence.

Also, IQL needs two hyperparameters ($\tau$ and $\alpha$) while SQL only needs one ($\alpha$). The two hyperparameters in IQL may not align well because they represent two different regularizations. If we set $f(x) = \log(x)$ in our IVR framework, we will find objective (3) is extracting the optimal policy that uses the Reverse KL divergence as the value regularization. However, the corresponding optimal

Table 1: Averaged normalized scores of SQL against other baselines. The scores are taken over the final 10 evaluations with 10 seeds. SQL achieved the highest scores in 12 out of 14 tasks.

| Dataset | BC | 10%BC | BCQ | DT | One-step | TD3+BC | CQL | IQL | SQL (ours) |
|---|---|---|---|---|---|---|---|---|---|
| halfcheetah-m | 42.6 | 42.5 | 47.0 | 42.6 | 48.4 | 48.3 | 44.0 | 47.4 | 48.3 ±0.2 |
| hopper-m | 52.9 | 56.9 | 56.7 | 67.6 | 59.6 | 59.3 | 58.5 | 66.3 | 73.5 ±1.4 |
| walker2d-m | 75.3 | 75.0 | 72.6 | 74.0 | 81.8 | 83.7 | 72.5 | 78.3 | 84.2 ±0.6 |
| halfcheetah-m | 36.6 | 40.6 | 40.4 | 36.6 | 38.1 | 44.6 | 45.5 | 44.2 | 44.8±0.7 |
| hopper-m-r | 18.1 | 75.9 | 53.3 | 82.7 | 97.5 | 60.9 | 95.0 | 94.7 | 96.7 ±5.3 |
| walker2d-m | 26.0 | 62.5 | 52.1 | 66.6 | 49.5 | 81.8 | 77.2 | 73.9 | 77.2±14.8 |
| halfcheetah-m-e | 55.2 | 92.9 | 89.1 | 86.8 | 93.4 | 90.7 | 91.6 | 86.7 | 94.0 ±0.4 |
| hopper-m-e | 52.5 | 110.9 | 81.8 | 107.6 | 103.3 | 98.0 | 105.4 | 91.5 | 111.8 ±0.2 |
| walker2d-m-e | 107.5 | 109.0 | 109.0 | 108.1 | 113.0 | 110.1 | 108.8 | 109.6 | 110.0±0.8 |
| MuJoCo mean | 51.9 | 74.0 | 66.9 | 74.8 | 76.1 | 75.2 | 77.6 | 76.9 | 82.3 ±2.7 |
| antmaze-u | 54.6 | 62.8 | 78.9 | 59.2 | 64.3 | 78.6 | 84.8 | 87.5 | 92.2 ±3.4 |
| antmaze-u-d | 45.6 | 50.2 | 55.0 | 53.0 | 60.7 | 71.4 | 43.4 | 62.2 | 78.0 ±2.3 |
| antmaze-m-p | 0 | 5.4 | 0 | 0.0 | 0.3 | 10.6 | 65.2 | 71.2 | 74.7 ±4.7 |
| antmaze-m-d | 0 | 9.8 | 0 | 0.0 | 0.0 | 3.0 | 54.0 | 70.0 | 75.0 ±4.2 |
| antmaze-l-p | 0 | 0.0 | 6.7 | 0.0 | 0.0 | 0.2 | 38.4 | 39.6 | 43.8 ±5.8 |
| antmaze-l-d | 0 | 6.0 | 2.2 | 0.0 | 0.0 | 0.0 | 31.6 | 47.5 | 49.8 ±7.7 |
| antmaze mean | 16.7 | 21.3 | 23.8 | 18.7 | 20.9 | 27.3 | 61.9 | 63.0 | 68.9 ±4.7 |
| kitchen-c | 33.8 | - | - | - | - | - | 43.8 | 61.4 | 76.4 ±8.7 |
| kitchen-p | 33.9 | - | - | - | - | - | 49.8 | 46.1 | 65.5 ±9.4 |
| kitchen-m | 47.5 | - | - | - | - | - | 51.0 | 52.8 | 61.4 ±5.4 |
| kitchen mean | 38.4 | - | - | - | - | - | 48.2 | 53.4 | 67.7 ±7.8 |

$V$-function learning objective is not objective (2). This reveals that the policy extraction part in IQL gets a different policy from the one implied in the optimal $Q$-function.

Like OptiDICE (Lee et al., 2021), SQL derives the solution by solving a regularized optimization problem. However, OptiDICE solves an upper bound of the learning objective while SQL solves the exact one. Additionally, OptiDICE solves for the state visitation distribution of the optimal policy, $d^*$, rather than $\pi^*$, this resutls a residual learning scheme, which is known to have slower convergence than fitted TD-learning (Baird, 1995).

# 5 EXPERIMENTS

We present empirical evaluations of SQL in this section. We first evaluate SQL against other baseline algorithms on D4RL (Fu et al., 2020) benchmark datasets. We then show the benefits of sparsity introduced in SQL by comparing SQL with IQL in noisy data regimes. We finally show the robustness of SQL by comparing SQL with CQL in small data regimes.

## 5.1 D4RL BENCHMARK DATASETS

We first evaluate our approach on D4RL MuJoCo, AntMaze, and Kitchen datasets (Fu et al., 2020). It is worth mentioning that Antmaze and Kitchen datasets include few or no near-optimal trajectories, and highly require learning a value function to obtain effective policies via "stitching".

We compare SQL with prior state-of-the-art offline RL methods, including BC (Sammut, 2010), 10%BC (Chen et al., 2021), BCQ (Fujimoto et al., 2018), DT(Chen et al., 2021), TD3+BC (Fujimoto & Gu, 2021), One-step RL (Brandfonbrener et al., 2021), CQL (Kumar et al., 2020), and IQL (Kostrikov et al., 2021a). The aggregated results are displayed in Table 1. In MuJoCo locomotion tasks, where performance is already saturated, SQL shows competitive results to the best performance of prior methods. While in more challenging AntMaze and kitchen tasks, SQL outperforms all other baselines by a large margin. This shows the effectiveness of value learning in SQL. Moreover, SQL convergences much faster than all prior methods, e.g., 0.2 million training steps for most tasks. Full experimental details and learning curves can be found in Appendix 8.

## 5.2 NOISY DATA REGIME

In this section, we try to validate our hypothesis that the sparsity term SQL introduced in learning the value function will benefit when the datasets contain a large portion of noisy transitions. To do so, we make a "mixed" dataset by combining the random dataset and the expert dataset with different expert ratios. We test the performance of SQL and IQL under different mixing ratios, as shown in Fig. 2.

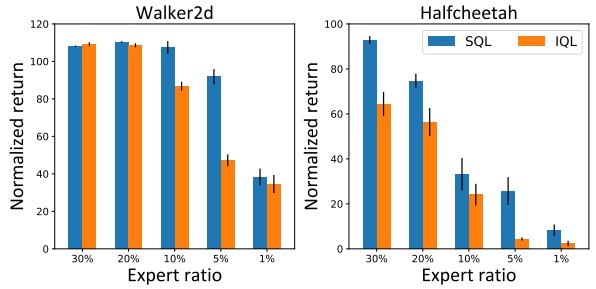

Figure 2: The performance of SQL vs IQL in noisy data regimes.

It is shown that SQL consistently outperforms IQL under all different expert ratios. The performance of IQL is vulnerable to the expert ratio, it has a sharp decrease from 30% to 1% while SQL could still remain the expert performance. For example, in walker2d, SQL reaches near 100 performance when the expert ratio is only 5%; in halfcheetah, IQL is affected even with a high expert ratio (30%).

## 5.3 SMALL DATA REGIME

In this section, we try to explore the benefits of in-sample learning over out-of-sample learning. We are interested to see whether in-sample learning brings more robustness than out-of-sample learning when the dataset size is small or the dataset diversity of some states is small, which are challenges one might encounter when using offline RL algorithms on real-world data.

To do so, we make custom datasets by discarding some transitions in the AntMaze datasets. For each transition, the closer it is to the target location, the higher probability it will be discarded from the dataset. This simulates the

Table 2: The normalized return (NR) and Bellman error (BR) of SQL vs CQL in small data regimes.

| Dataset (Antmaze) | | CQL | | SQL | |
|---|---|---|---|---|---|
| | | NR | BE | NR | BE |
| Medium | Vanilla | 65.2 | 13.1 | 70.0 | 1.6 |
| | Easy | 48.2 | 14.8 | 56.2 | 1.7 |
| | Medium | 14.5 | 14.7 | 43.3 | 2.1 |
| | Hard | 9.3 | 64.4 | 24.2 | 1.9 |
| Large | Vanilla | 38.4 | 13.5 | 49.8 | 1.4 |
| | Easy | 28.1 | 12.8 | 40.5 | 1.5 |
| | Medium | 6.3 | 30.6 | 36.7 | 1.3 |
| | Hard | 0 | 300.5 | 34.2 | 2.6 |

situation where the dataset is fewer and has limited state coverage near the target location because the data generation policies maybe not be satisfied and are more determined when they get closer to the target location (Kumar et al., 2022). We use a hyperparameter to control the discarding ratio and build three new tasks: `Easy`, `Medium` and `Hard`, with dataset becomes smaller. For details please refer to Appendix 8. We compare SQL with CQL as they use the same inherent value regularization but SQL uses in-sample learning while CQL uses out-of-sample learning.

We demonstrate the final normalized return (NR) during evaluation and the mean squared Bellman error (BE) during training in Table 2. It is shown that CQL has a significant performance drop when the difficulty of tasks increases, the Bellman error also exponentially grows up, indicating that the value extrapolation error becomes large in small data regimes. SQL remains a stable yet good performance under all difficulties, the Bellman error of SQL is much smaller than that of CQL. This justifies the benefits of in-sample learning, i.e., it avoids erroneous value estimation by using only dataset samples while still allows in-sample generalization to obtain a good performance.

## 6 CONCLUSIONS AND FUTURE WORK

In this paper, we propose a general Implicit Value Regularization framework, which builds the bridge between behavior regularized and in-sample learning methods in offline RL. Based on this framework, we propose a practical algorithm, which uses the same value regularization as CQL, but in a complete in-sample manner. We verify the effectiveness of SQL on D4RL datasets. We also show the robustness of SQL in noisy and small data regimes by comparing it with different baselines. One future work is to use other choices of $f$, for example $\log(x)$, this brings a different algorithm that no longer introduces sparsity. Another future work is, instead of only constraining action distribution, constraining the state-action distribution between $d^\pi$ and $d^D$ as considered in Nachum et al. (2019).

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

## 7 PROOFS

### 7.1 PROOF OF THEOREM 1

In this section, we give the detailed proof for Theorem 1, which states the optimality condition of the behavior regularized MDP. The proof follows from the Karush-Kuhn-Tucker (KKT) conditions where the derivative of a Lagrangian objective function with respect to policy $\pi(a|s)$ is set zero. Hence, our main theory is necessary and sufficient.

*Proof.* The Lagrangian function of (4) obtained by the optimal policy is written as follows

$$L(\pi, \beta, u) = \sum_s d_\pi(s) \sum_a \pi(a|s) \left( Q^*(s, a) - \alpha f \left( \frac{\pi(a|s)}{\mu(a|s)} \right) \right)$$
$$- \sum_s d_\pi(s) \left[ u(s) \left( \sum_a \pi(a|s) - 1 \right) + \sum_a \beta(a|s) \pi(a|s) \right],$$

where $d_\pi$ is the stationary state distribution of the policy $\pi$, $\mu$ and $\beta$ are Lagrangian multipliers for the equality and inequality constraints respectively.

Let $h_f(x) = xf(x)$. Then the KKT condition of (4) are as follows, for all states and actions we have

$$0 \le \pi(a|s) \le 1 \text{ and } \sum_a \pi(a|s) = 1 \tag{15}$$

$$0 \le \beta(a|s) \tag{16}$$
$$\beta(a|s)\pi(a|s) = 0 \tag{17}$$
$$Q^*(s, a) - \alpha h'_f \left( \frac{\pi(a|s)}{\mu(a|s)} \right) - u(s) + \beta(a|s) = 0 \tag{18}$$

where (15) is the feasibility of the primal problem, (16) is the feasibility of the dual problem, (17) results from the complementary slackness and (18) is the stationarity condition. We eliminate $d_\pi(s)$ since we assume all policies induce an irreducible Markov chain.

From (18), we can resolve $\pi(a|s)$ as

$$\pi(a|s) = \mu(a|s) \cdot g_f \left( \frac{1}{\alpha} (Q^*(s, a) - u(s) + \beta(a|s)) \right)$$

Fix a state $s$. For any positive action, its corresponding Lagrangian multiplier $\beta(a|s)$ is zero due to the complementary slackness and $Q^*(s, a) > u(s) + \alpha h'_f(0)$ must hold. For any zero-probability action, its Lagrangian multiplier $\beta(a|s)$ will be set such that $\pi(a|s) = 0$. Note that $\beta(a|s) \ge 0$, thus $Q^*(s, a) \le u(s) + \alpha h'_f(0)$ must hold in this case. From these observations, $\pi(a|s)$ can be reformulated as

$$\pi(a|s) = \max \left\{ g_f \left( \frac{1}{\alpha} (Q^*(s, a) - u(s)) \right), 0 \right\} \tag{19}$$

Next we aim to obtain the optimal state value $V^*$. It follows that

$$V^*(s) = \mathcal{T}_f^* V^*(s)$$
$$= \sum_a \pi^*(a|s) \left( Q^*(s, a) - \alpha f \left( \frac{\pi^*(a|s)}{\mu(a|s)} \right) \right)$$
$$= \sum_a \pi^*(a|s) \left( u^*(s) + \alpha \frac{\pi^*(a|s)}{\mu(a|s)} f' \left( \frac{\pi^*(a|s)}{\mu(a|s)} \right) \right)$$
$$= u^*(s) + \alpha \sum_a \frac{\pi^*(a|s)^2}{\mu(a|s)} f' \left( \frac{\pi^*(a|s)}{\mu(a|s)} \right)$$
$$= u^*(s) + \alpha \mathbb{E}_{a \sim \mu} \left[ \left( \frac{\pi^*(a|s)}{\mu(a|s)} \right)^2 f' \left( \frac{\pi^*(a|s)}{\mu(a|s)} \right) \right]$$

The first equality follows from the definition of the optimal state value. The second equality holds because $\pi^*$ maximizes $\mathcal{T}_f^* V^*(s)$. The third equality results from plugging (18).

To summarize, we obtain the optimality condition of the behavior regularized MDP as follows

$$Q^*(s,a) = r(s,a) + \gamma \mathbb{E}_{s'|s,a} \left[ V^*(s') \right]$$

$$\pi^*(a|s) = \mu(a|s) \cdot \max \left\{ g_f \left( \frac{Q^*(s,a) - u^*(s)}{\alpha} \right), 0 \right\}$$

$$V^*(s) = u^*(s) + \alpha \mathbb{E}_{a \sim \mu} \left[ \left( \frac{\pi^*(a|s)}{\mu(a|s)} \right)^2 f' \left( \frac{\pi^*(a|s)}{\mu(a|s)} \right) \right]$$

$\square$

## 7.2 PROOF OF THEOREM 2

*Proof.* For any two state value functions $V_1$ and $V_2$, let $\pi_i$ be the policy that maximizes $\mathcal{T}_f^* V_i$, $i \in 1, 2$. Then it follows that for any state s in $\mathcal{S}$,

$$\left( \mathcal{T}_f^* V_1 \right)(s) - \left( \mathcal{T}_f^* V_2 \right)(s)$$

$$= \sum_a \pi_1(a|s) \left[ r + \gamma \mathbb{E}_{s'} \left[ V_1(s') \right] - \alpha f \left( \frac{\pi_1(a|s)}{\mu(a|s)} \right) \right] - \max_\pi \sum_a \pi(a|s) \left[ r + \gamma \mathbb{E}_{s'} \left[ V_2(s') \right] - \alpha f \left( \frac{\pi(a|s)}{\mu(a|s)} \right) \right]$$

$$\leq \sum_a \pi_1(a|s) \left[ r + \gamma \mathbb{E}_{s'} \left[ V_1(s') \right] - \alpha f \left( \frac{\pi_1(a|s)}{\mu(a|s)} \right) \right] - \sum_a \pi_1(a|s) \left[ r + \gamma \mathbb{E}_{s'} \left[ V_2(s') \right] - \alpha f \left( \frac{\pi_1(a|s)}{\mu(a|s)} \right) \right]$$

$$= \gamma \sum_a \pi_1(a|s) \mathbb{E}_{s'} \left[ V_1(s') - V_2(s') \right] \leq \gamma \left\| V_1 - V_2 \right\|_\infty$$

By symmetry, it follows that for any state $s$ in $\mathcal{S}$,

$$\left( \mathcal{T}_f^* V_1 \right)(s) - \left( \mathcal{T}_f^* V_2 \right)(s) \leq \gamma \left\| V_1 - V_2 \right\|_\infty$$

Therefore, it follows that

$$\left\| \mathcal{T}_f^* V_1 - \mathcal{T}_f^* V_2 \right\|_\infty \leq \gamma \left\| V_1 - V_2 \right\|_\infty$$

$\square$

## 8 EXPERIMENTAL DETAILS

**D4RL experimental details** Our implementation of 10%BC is as follows, we first filter the top 10 % trajectories in terms of the trajectory return, and then run behaviour cloning on those filtered data. We re-run CQL on Antmaze datasets as we find the performance can be improved by carefully sweeping the hyperparameter `min-q-weight`, using the PyTorch-version implementation[2]. Other baseline results are taken directly from their corresponding papers. For the results of IQL on Kitchen tasks, we follow the author's instructions and re-run the author-provided implementation[3].

For MuJoCo locomotion and Kitchen tasks, we average mean returns over 10 evaluations every 5000 training steps, over 10 random seeds. For AntMaze tasks, we average over 100 evaluations every 0.1M training steps, over 10 random seeds. Followed by IQL, we standardize the rewards by dividing the difference in returns of the best and worst trajectories in MuJoCo and kitchen tasks, we subtract 1 to rewards in Antmaze tasks.

In SQL, we use 2-layer MLP with 256 hidden units, we use Adam optimizer Kingma & Ba (2015) with a learning rate of $3 \cdot 10^{-4}$ for all neural networks. Following Mnih et al. (2013); Lillicrap et al. (2016), we introduce a target critic network with soft update weight $5 \cdot 10^{-3}$. We implement our method in the framework of JAX Bradbury et al. (2018). For the only hyperparameter $\alpha$, we sweep it

---

[2]https://github.com/young-geng/CQL
[3]https://github.com/ikostrikov/implicit_q_learning

from $[0.2, 0.5, 1, 2, 5, 10]$ for all tasks and choose the best one as our reported scores, as well as in other experimental settings.

**Nosiy data regime experimental details**    In this experiment setting, we introduce the `noisy` dataset by mixing the `expert` and `random` dataset with different expert using MuJoCo locomotion datasets. The number of total transitions of the noisy dataset is $100,000$. We provide details in Table 3. We report the score of IQL by choosing the best score from $\tau$ in $[0.5, 0.6, 0.7, 0.8, 0.9]$.

Table 3: Noisy dataset of MuJoCo locomotion tasks with different expert ratios.

| Env | Expert ratio | Total transitions | Expert transitions | Random transitions |
|---|---|---|---|---|
| | 1% | 100,000 | 1,000 | 99,000 |
| | 5% | 100,000 | 5,000 | 95,000 |
| Walker2d | 10% | 100,000 | 10,000 | 90,000 |
| | 20% | 100,000 | 20,000 | 80,000 |
| | 30% | 100,000 | 30,000 | 70,000 |
| | 1% | 100,000 | 1,000 | 99,000 |
| | 5% | 100,000 | 5,000 | 95,000 |
| Halfcheetah | 10% | 100,000 | 10,000 | 90,000 |
| | 20% | 100,000 | 20,000 | 80,000 |

**Small data regime experimental details**    We generate the small dataset using the following psedocode 1, its hardness level can be found at Table 4. We report the score of CQL by choosing the best score from `min-q-weight` in $[0.5, 1, 2, 5, 10]$.

Listing 1: The sketch of generation procedure of small data regimes with different hard levels. Given an Antmaze environment and a hardness level, we discard some transitions by following the rule in the Coding List. Intuitively, the closer the transition is to the `GOAL`, the higher the probability that it will be discarded.

```
# hardness = {'easy': easy, 'medium': medium, 'hard': hard}
obs = dataset['observations']
length = dataset['observations'].shape[0]
POSITIONS = env.get_position(obs)
GOAL = env.get_goal()
MINIMAL_POSITION = env.get_minimal_position()
# get maximal Euclidean distance
MAX_EU_DIS = (GOAL - MINIMAL_POSITION)**2
DIS = ((POSITIONS MINIMAL_POSITION)**2) / MAX_EU_DIS
save_idx = np.random.random(size=length) > (DIS +
    hardness['LEVEL_OF_HARD'])
small_data = collections.defaultdict()
for key in dataset.keys():
    small_data[key] = dataset[key][save_idx]
```

Table 4: Details of small data regimes with different task difficulties.

| Dataset (AntMaze) | | Hardness | Total transitions | Reward signals |
|---|---|---|---|---|
| | Vanilla | NA | 100, 000 | 10,000 |
| medium-play | Easy | 0 | 56,000 | 800 |
| | Medium | 0.07 | 48,000 | 150 |
| | Hard | 0.1 | 45,000 | 10 |
| | Vanilla | NA | 100,000 | 12500 |
| | Easy | 0 | 72,000 | 5,000 |
| large-play | Medium | 0.3 | 42,000 | 1,000 |
| | Medium | 0.35 | 37,000 | 500 |
| | Hard | 0.38 | 35,000 | 100 |

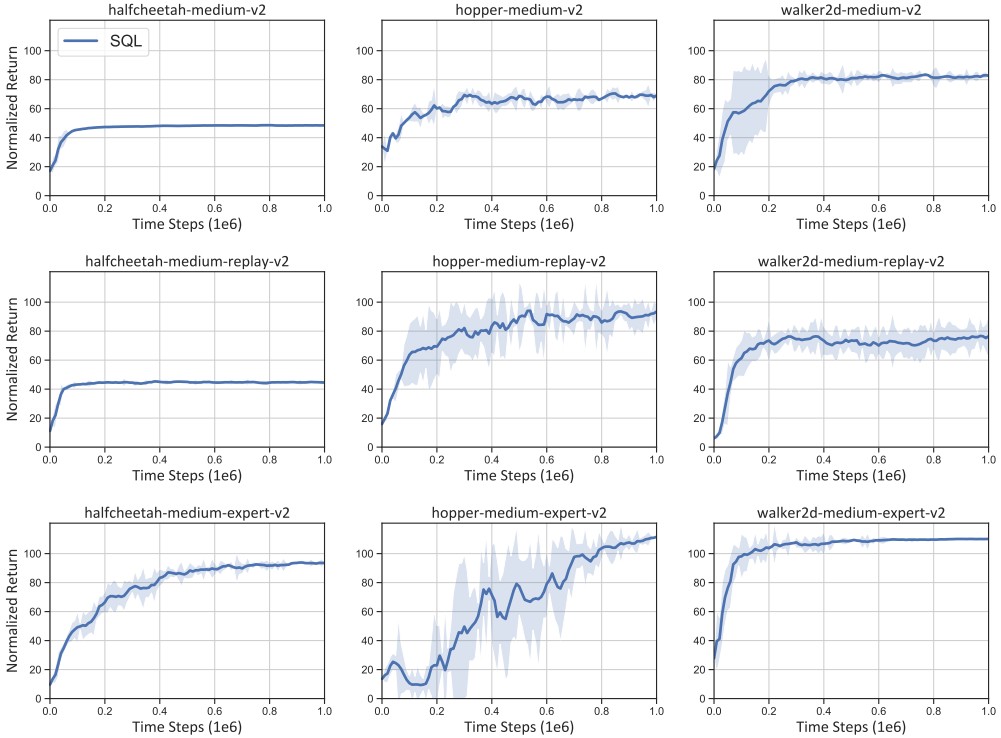

Figure 3: Learning curves of SQL on MuJoCo locomotion datasets.

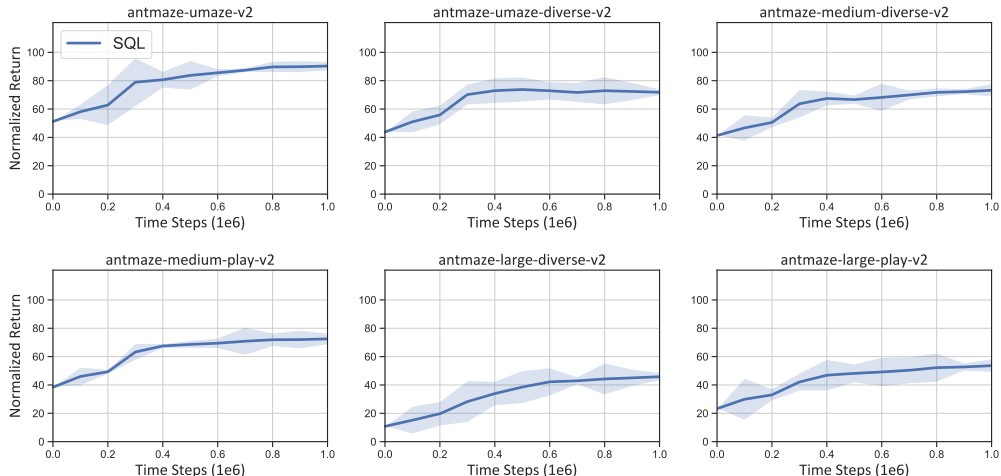

Figure 4: Learning curves of SQL on AntMaze datasets.

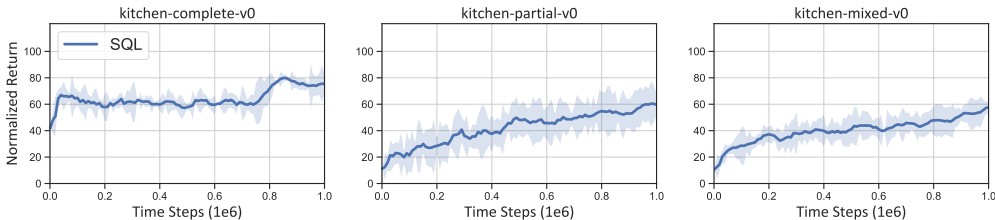

Figure 5: Learning curves of SQL on Kitchen datasets.

