# OpenReview forum: "Sparse Q-Learning: Offline Reinforcement Learning with Implicit Value Regularization"
_NeurIPS.cc/2022/Workshop/Offline_RL — Offline RL Workshop NeurIPS 2022_

### Official Review · Reviewer_Y5Sy · 2022-10-07

**Rating:** 6
**Confidence:** 2

**Review:**

This paper makes a key finding that the in-sample learning paradigm arises under the Implicit Value Regularization (IVR) framework. This gives a deeper understanding of why the in-sample learning paradigm works, i.e., it applies implicit value regularization to the policy. Based on the IVR framework, we further propose a practical algorithm, which uses the same value regularization as CQL, but in a complete in-sample manner. Compared with IQL, we find that our algorithm introduces sparsity in learning the value function, we thus dub our method Sparse Q-learning (SQL).

This paper verifies the effectiveness of SQL on D4RL benchmark datasets. We also show the benefits of sparsity by comparing SQL with IQL in noisy data regimes and show the robustness of in- sample learning by comparing SQL with CQL in small data regimes.

Question: the benefit of sparsity is not very clear, for example, sparsity may remind people that you are using sparse to do offline learning. Also, the total score in the table is not reported.

---

### Official Review · Reviewer_sg4y · 2022-10-19

**Rating:** 6
**Confidence:** 4

**Review:**

This paper proposes an in-sample based offline RL algorithm, Sparse Q-Learning (SQL), based on the implicit value regularization principle. The method is derived from a regularized MDP formulation and bears interesting connection to existing works such as CQL and IQL. SQL outperforms prior algorithms on the standard D4RL benchmark, and is superior in the noisy and small data regime.

This is a sound paper. I did not check the mathematical derivation carefully, but they look reasonable. The empirical improvement is relatively small. In terms of related work, it may be good to compare to (algorithmically) and reference some prior work that also does in-sample learning and derive offline RL algorithms from f-divergence and duality. For example, https://arxiv.org/abs/2206.03023 and https://openreview.net/forum?id=BrPdX1bDZkQ.